# Flexible adaptation of task-positive brain networks predicts efficiency of evidence accumulation
Alexander Weigard [1] ✉, Mike Angstadt[1], Aman Taxali[1], Andrew Heathcote[2], Mary M. Heitzeg[1] & Chandra Sripada [1]

Efficiency of evidence accumulation (EEA), an individual's ability to selectively gather goal-relevant information to make adaptive choices, is thought to be a key neurocomputational mechanism associated with cognitive functioning and transdiagnostic risk for psychopathology. However, the neural basis of individual differences in EEA is poorly understood, especially regarding the role of largescale brain network dynamics. We leverage data from 5198 participants from the Human Connectome Project and Adolescent Brain Cognitive Development Study to demonstrate a strong association between EEA and flexible adaptation to cognitive demand in the "task-positive" frontoparietal and dorsal attention networks. Notably, individuals with higher EEA displayed divergent task-positive network activation across n-back task conditions: higher activation under high cognitive demand (2-back) and lower activation under low demand (0-back). These findings suggest that brain networks' flexible adaptation to cognitive demands is a key neural underpinning of EEA.

Evidence accumulation models[1] posit that individuals complete many cognitive tasks by gradually accumulating noisy evidence for each possible choice until evidence for one choice reaches a critical threshold. This class of formal models has been highly successful at explaining key features of choice response time data and is now considered one of the predominant mathematical frameworks for modeling task performance across a wide variety of cognitive domains in the psychological and neural sciences[1–3].

A growing literature has recently begun to reveal how the latent psychological mechanisms posited by evidence accumulation models contribute to higher-order cognition and behavior. Efficiency of evidence accumulation (EEA), or the ability to selectively accumulate goal-relevant evidence to make adaptive choices in the context of noisy information, appears to be a task-general process and a key underpinning of higher-order cognitive functions, including working memory and general intelligence[4–10]. In parallel, recent applications of these models in clinical research have identified reduced EEA across multiple disorders, including attention-deficit/hyperactivity disorder (ADHD)[11–14], schizophrenia[15,16], bipolar disorder[17], and problematic substance use[18], suggesting that lower EEA is a transdiagnostic cognitive risk factor for psychopathology[17,19].

EEA's relevance to cognitive functioning and psychiatric disorders has led to increasing interest in identifying its neural underpinnings. Experimental work in non-human primates has focused on recordings of neural firing rates during oculomotor perceptual decision-making tasks, in which primates make saccadic responses that reflect, for example, the direction of coherent dot motion or the detection of oddball stimuli. Neural firing rates within regions linked to perceptual processing and oculomotor responding display ramping patterns that have similar properties to the decision process assumed by evidence accumulation models; longer latencies and slower buildup rates of these ramping patterns are linked with greater task difficulty and slower behavioral response times, as would be expected if they track evidence accumulation[2,20–24]. More recently, simultaneous modeling of behavioral and neural data within a single formal model has provided stronger quantitative evidence that these ramping patterns reflect the same evidence accumulation processes that describe behavioral saccade and response time data[25].

Parallel neural signatures of evidence accumulation have been identified in humans using electroencephalogram (EEG)[26–31] and functional magnetic resonance imaging (fMRI)[32–35]. Although multiple brain areas appear to be involved, converging evidence from experimental studies suggests that the frontoparietal network (FPN), a group of brain regions previously associated with task performance and cognitive control, plays a central role[36]. Outside of this experimental literature, less is known about the neural basis of individual variability in EEA. Recent studies using disparate methodologies have found that better EEA is associated with activation in the inferior parietal lobe during decision-making[37], greater error-related activations in the salience network[38], and a marker of neural speed derived from several EEG components[39]. Additionally, a recent multimodal

[1]Department of Psychiatry, University of Michigan, Ann Arbor, USA. [2]Department of Psychological Methods, University of Amsterdam, Amsterdam, Netherlands.
✉e-mail: asweigar@med.umich.edu

neuroimaging investigation that focused on the dorsal portion of the FPN found evidence that white matter macrostructure within this subnetwork and its functional coupling with premotor cortex were both related to EEA[40].

Despite the critical importance of this body of work for understanding the neural basis of EEA, a key limitation is that each of these studies had a constrained focus on specific brain regions or narrow subnetworks of regions. Recent research leveraging multivariate predictive modeling in neuroimaging data has demonstrated that many cognitive and psychological variables are only weakly associated with activity in discrete regions and are more robustly predicted by features of largescale brain networks that are distributed across the cortex[41–43]. Further, the properties of such brain networks are far from static. Instead, these networks display dynamic adaptations and reconfigurations to meet task demands[44,45]. Hence, although there is growing evidence that EEA may be a key neurocomputational underpinning of cognitive and adaptive functioning, its associations with the dynamic properties of largescale brain networks remain unclear.

In the current study, we present novel evidence that one such property shows a strong and robust association with EEA: the degree to which "task-positive" brain networks flexibly adapt to cognitive demand. The FPN and the dorsal attention network (DAN), another group of brain regions associated with the top-down control of attention[46], are collectively labeled "task-positive" networks[47] because they reliably show increased activity in task conditions that are cognitively demanding (i.e., difficult). As EEA is a formal measure of the ratio of task-relevant signal to task-irrelevant noise during cognitive processing[48,49], it is conceptually linked to the interrelated functions of the FPN, which appears to selectively facilitate goal-relevant behaviors during task performance, and the DAN, which appears to modulate attentional resources toward goal-relevant information.

Parametric effects of cognitive demand on activity in the FPN and DAN are reliably observable during the commonly used n-back fMRI paradigm[50,51], in which the difficulty of the cognitive task varies as a function of how many stimuli must be actively maintained in working memory to make accurate choices. Previous work has shown that higher levels of difficultly on the n-back generate neural activation maps that are more closely associated with cognitive abilities than those generated from less difficult n-back conditions[52], suggesting that the degree to which individuals' brain networks respond to the demands of a given task may have important implications for task performance. As flexible adaptation of neural systems to the demands of external tasks has long been theorized to support efficient cognitive processing[48,53], we sought to directly assess the degree to which demand-related changes in neural activation across the FPN and DAN are associated with EEA.

Across two large data sets spanning different developmental periods, the Human Connectome Project (HCP)[54] and the baseline sample of 9- and 10-year-old youth from Adolescent Brain Cognitive Development[SM] Study (ABCD Study[®])[55], we first use multivariate predictive modeling to demonstrate that neural response to cognitive demand during the n-back explains a substantial portion (36–39%) of the variance in individuals' EEA on the task. We then show that this predictive relationship can be largely attributed to EEA's association with demand-related activation patterns in the FPN and DAN. Critically, we provide novel evidence that this network configuration shows divergent relations with EEA under different levels of cognitive demand; although activation in task-positive networks during the difficult (2-back) condition is positively related to EEA, activation in these networks during the easy (0-back) condition is *negatively* related to EEA. These findings suggest that flexible adaptation to cognitive demands across task-positive brain networks is a key neural underpinning of EEA and its downstream consequences for cognition and behavior.

## Results
### Neural responses to cognitive demand during the n-back explain a sizable proportion of the variance in individuals' EEA on the task
We built multivariate models that used vertex-wise brain activation data from the n-back's cognitive load (2-0) contrast to predict EEA metrics during the n-back task (see Methods for details on EEA metrics). We tested their generalizability in independent data using leave-one-site-out cross-

validation[42] in ABCD and 10-fold cross-validation in HCP (as the HCP data were collected at a single study site). All analyses were adjusted for age, sex, race/ethnicity, and motion (framewise displacement) using the partial correlation technique described in Methods.

Neural responses to cognitive demand explained a large proportion of the variance across all measures of EEA in both samples (Fig. 1a) and performance was consistently high across all ABCD sites and HCP cross-validation folds (Fig. 1b). Performance of the models was highest when predicting the average of EEA across both n-back load conditions, explaining 39% of the variance in ABCD and 36% of the variance in HCP. Predictions of EEA on the 0-back (ABCD = 32%, HCP = 35%) were slightly more accurate than predictions of EEA on the 2-back condition (ABCD = 30%, HCP = 26%).

This general pattern indicates that neural responses to cognitive demand are strongly related to measures of EEA across both levels of n-back load. Combined with the large observed correlations between EEA measured on the 0- and 2-back tasks (ABCD $r = 0.45$, CI = 0.42–0.48; HCP $r = 0.54$, CI = 0.48–0.59), these results are consistent with the hypothesis that EEA reflects a domain-general latent process that drives performance across tasks of both low and high complexity and has common neural underpinnings regardless of specific task demands[19].

### Features predictive of EEA show substantial overlap with the task-positive network regions activated in the n-back's standard cognitive load contrast
Brain-wide consensus maps, which indicate the relative importance of activation from each cortical surface area for predicting EEA, were generated with feature weights from the models predicting average EEA with activation in the cognitive load (2-0) contrast. These consensus maps showed a strong visual similarity to the 2-0 contrast's group-average activation maps (Fig. 2). As expected, regions in the FPN and DAN were heavily represented across both types of maps. Most of the prefrontal and midline regions strongly activated by the load contrast were also heavily featured in the predictive model, although there were some apparent differences between the maps in lateral parietal regions. These spatial patterns were remarkably consistent across the ABCD and HCP samples. However, one notable difference between the samples is the finding of generally lower effect sizes in the group-average 2-0 activation map in ABCD relative to HCP, which could indicate that, although children and adults activate similar networks during high cognitive demand, activation levels are generally lower in children compared to adults, consistent with recent findings[56].

### Activations in task-positive networks and the somatomotor network make substantial contributions to prediction of EEA
To further parse out the role of specific largescale brain networks in predicting EEA, we examined associations between EEA and load-related differences (2-0) in the activation of all seven Yeo networks (averaged across the entire networks' parcellations) for both ABCD and HCP (Table 1, top panel). As expected, load-related differences in the FPN and DAN consistently showed moderate- to large-sized positive relations with EEA, suggesting that neural responses to cognitive demand in both task-positive networks make key contributions to the fMRI data's predictive associations. Of the five remaining networks, three (VAN, VIS, and DMN) showed associations with EEA that were relatively small and were either not consistently significant across the two samples or were in inconsistent directions. Unexpectedly, we also found that load-related activation differences in the somatomotor (SMN) and limbic (LIM) networks were both consistently negatively related to EEA. The size of the SMN association suggested a comparable contribution to those of task-positive networks (absolute values of 95% CIs greater than, or largely overlapping with, those of the DAN) while the size of the LIM association was substantially smaller.

To quantify the proportion of the association between load-related activation and EEA that can be attributed to just the FPN, DAN and SMN, we fit multiple regression models in which only these three networks'

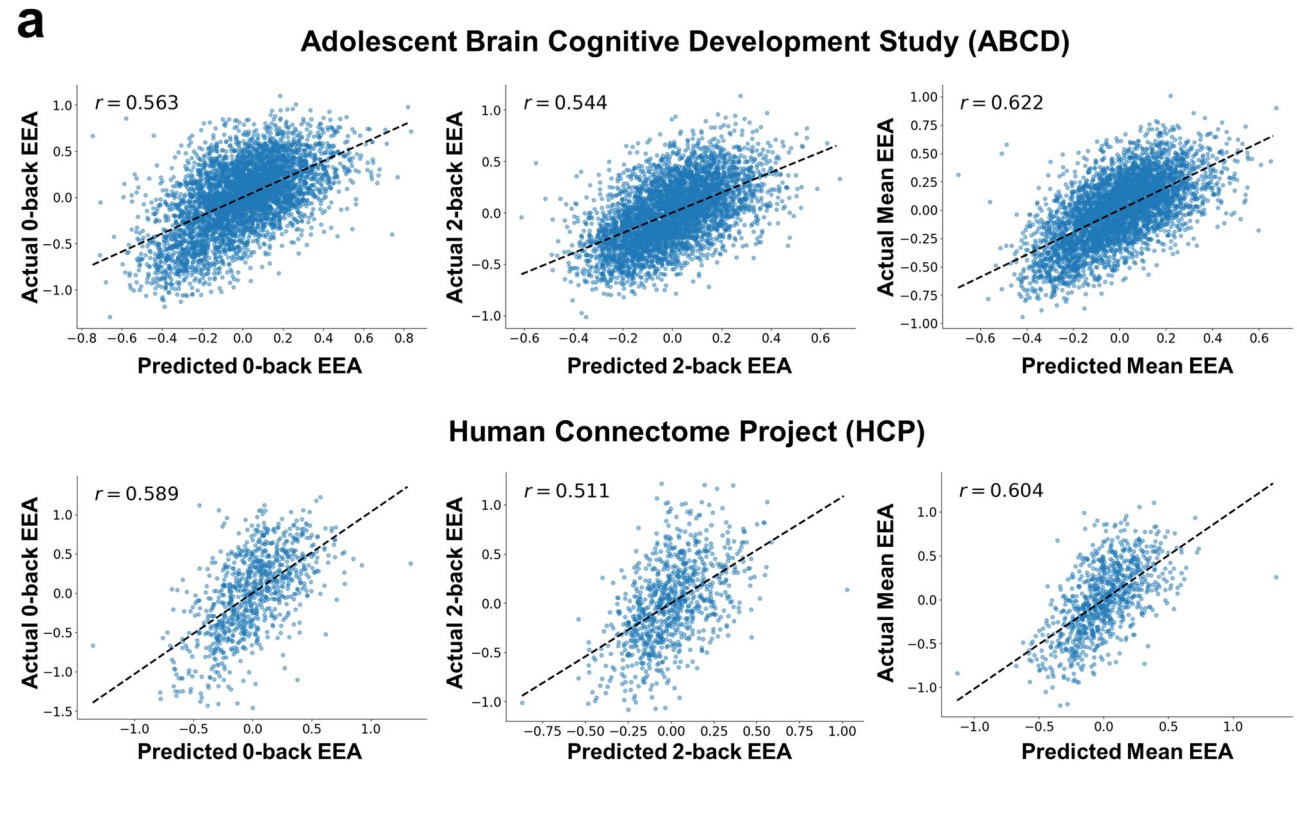

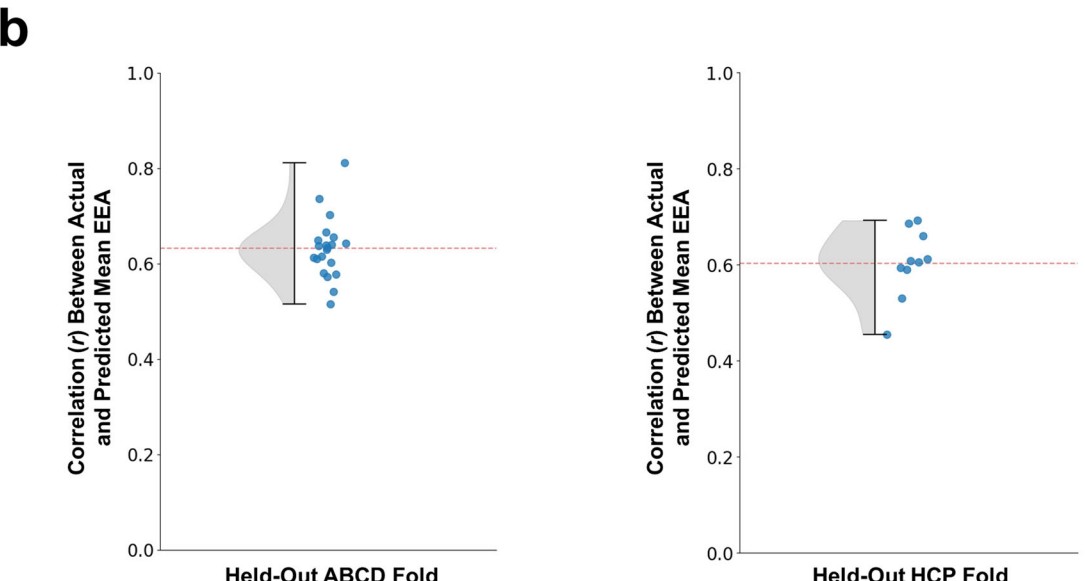

**Fig. 1 | Relations between brain activation in the n-back cognitive load (2-0) contrast and efficiency of evidence accumulation (EEA). a** Correlations between EEA values predicted by the multivariate model and actual EEA values for the 0-back task, 2-back task, and the mean across both tasks in the Adolescent Brain Cognitive Development Study (ABCD) and Human Connectome Project (HCP) samples. The predicted values are drawn from models fit to independent data using the leave-one-site-out and 10-fold cross-validation methods in the ABCD and HCP samples, respectively. All values are residuals from regressions that were adjusted for age, sex, race/ethnicity, and motion covariates. **b** Correlations between predicted and actual mean EEA values in each of the 10 HCP test folds and each of the ABCD sites. The density plot represents the distribution of values and the red line represents the average value.

average activations predicted EEA and compared the size of the association between observed EEA and model-predicted EEA with the size of the same association for a multiple regression model using average activations from all seven networks. The performance of the three-network regression model was only slightly lower than that of the full seven-network regression model in ABCD (three-network $r = 0.47$, CI = 0.44–0.50; full $r = 0.51$, CI = 0.49–0.54) and the 95% CIs of the two regression models largely overlapped in HCP (three-network $r = 0.52$, CI = 0.47–0.57; full $r = 0.55$,

**Fig. 2 | Group-level cortical maps of effect sizes (Cohen's d) from the n-back cognitive load (2-0) contrast and feature weights (converted to Z-scores: mean = 0, SD = 1) from the models predicting individuals' n-back task performance with this contrast. a** Effect size map for the 2-0 contrast in the Adolescent Brain Cognitive Development Study (ABCD) sample. **b** Effect size map for the 2-0 contrast in the Human Connectome Project (HCP) sample. **c** Consensus feature weight map for models predicting performance in the ABCD sample. **d** Consensus feature weight map for models predicting performance in the HCP sample. Source data for all maps are publicly available at: https://figshare.com/s/ab78c31c258e5c3e36b3.

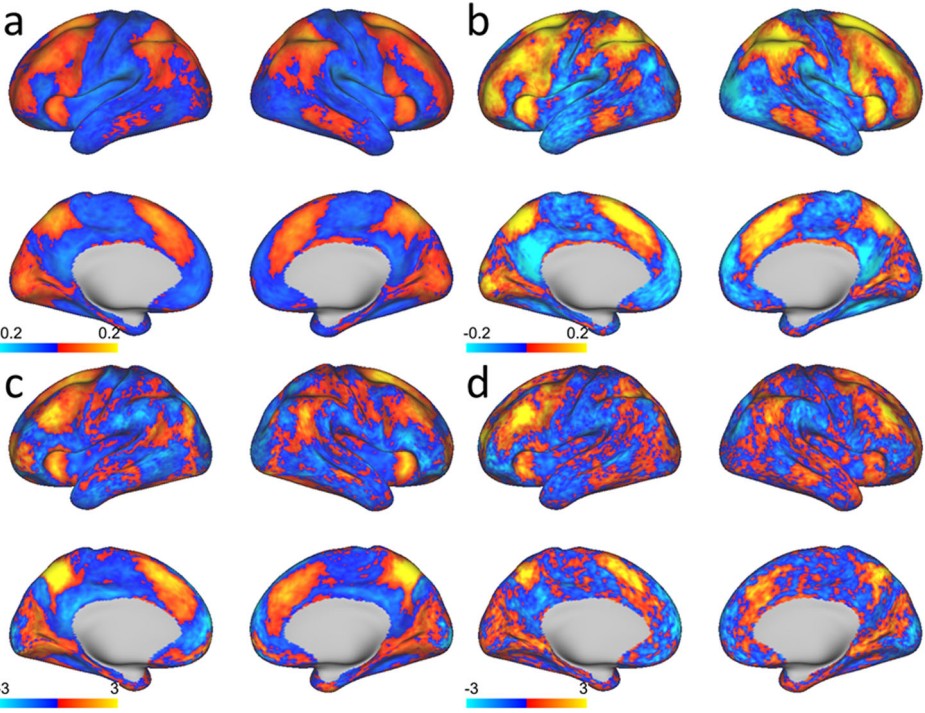

CI = 0.51–0.60). Therefore, the vast majority of the association between load-related neural activation and EEA appears attributable to the two task-positive networks of interest (FPN and DAN) and the SMN. We therefore focused all subsequent analyses on these three networks.

## Table 1 | Adolescent Brain Cognitive Development Study (ABCD) and Human Connectome Project (HCP) correlations between efficiency of evidence accumulation (EEA) and whole-network average measures of activation in the cognitive load (2-0) contrast (top panel), as well as correlations of EEA with 0-back and 2-back activation, relative to baseline, for the networks with the strongest associations (bottom panel)

| | ABCD r | ABCD 95% CI | | HCP r | HCP 95% CI | |
|---|---|---|---|---|---|---|
| FPN 2-0 | 0.36 | 0.33 | 0.39 | 0.46 | 0.41 | 0.51 |
| DAN 2-0 | 0.21 | 0.19 | 0.24 | 0.30 | 0.23 | 0.35 |
| VIS 2-0 | −0.09 | −0.07 | −0.12 | 0.07 | 0.00 | 0.13 |
| SMN 2-0 | −0.30 | −0.27 | −0.32 | −0.27 | −0.21 | −0.33 |
| VAN 2-0 | −0.07 | −0.03 | −0.10 | −0.01 | −0.08 | 0.06 |
| LIM 2-0 | −0.17 | −0.15 | −0.19 | −0.12 | −0.05 | −0.19 |
| DMN 2-0 | 0.01 | −0.01 | 0.03 | −0.01 | −0.08 | 0.06 |
| FPN 0 | −0.27 | −0.24 | −0.29 | −0.31 | −0.24 | −0.37 |
| FPN 2 | 0.17 | 0.13 | 0.22 | 0.08 | 0.02 | 0.15 |
| DAN 0 | −0.07 | −0.04 | −0.09 | −0.14 | −0.06 | −0.21 |
| DAN 2 | 0.17 | 0.14 | 0.21 | 0.08 | 0.00 | 0.15 |
| SMN 0 | 0.02 | −0.01 | 0.06 | 0.12 | 0.05 | 0.18 |
| SMN 2 | −0.29 | −0.26 | −0.32 | −0.06 | −0.13 | 0.00 |

All variables were adjusted for age, sex, race/ethnicity, and motion (framewise displacement) by using multiple regression to remove variance associated with these covariates. 95% confidence intervals, displayed in italics next to each correlation, were estimated using a clustered bootstrapping procedure that accounted for nesting by family and study site.
*FPN* frontoparietal network, *DAN* dorsal attention network, *VIS* visual network, *SMN* somatomotor network, *VAN* ventral attention network, *LIM* limbic network, *DMN* default mode network.

## Task-positive network activations during the 2-back and 0-back are positively correlated with one another but show strongly divergent relations with EEA

For the task-positive networks of interest and for the SMN, there were strong positive correlations between an individual's activation in the 2-back condition and their activation in the 0-back condition, both in the HCP sample (FPN *r* = 0.64, CI = 0.60–0.68; DAN *r* = 0.74, CI = 0.71–0.77; SMN *r* = 0.72, CI = 0.70–0.75) and in the ABCD sample (FPN *r* = 0.24, CI = 0.21–0.26; DAN *r* = 0.36, CI = 0.33–0.38; SMN *r* = 0.38, 0.35–0.40). These strong dependencies were notable given that task-positive network activation in the 2-back and 0-back conditions showed strongly *divergent* relationships with EEA (Table 1, bottom panel). More specifically, 2-back FPN and DAN network activation is positively related to EEA while 0-back activation in the same networks is negatively related to EEA.

For the SMN, trends in the opposite directions emerged; SMN activation during the 2-back was *negatively* related to EEA, although this association was only reliably different from 0 in the ABCD study, while SMN activation in the 0-back condition was positively related to EEA in HCP only.

Figure 3 illustrates these complex interrelations by plotting each individual's 0-back activation levels on the x-axis against their 2-back activation levels on the y-axis. The strong, positive relationship between 2-back and 0-back activation for all three networks is shown by the black dashed regression line. For both task-positive networks (FPN and DAN) in both the ABCD and HCP samples, we observed a common pattern. Individuals in the upper left-hand quadrant, who have relatively greater activation in the 2-back condition than would be expected given their lower activation in the 0-back condition, show the highest EEA, which is indicated both by the darker red hue of the points as well as the mean standardized EEA scores displayed in the quadrant. Individuals in the lower right-hand quadrant, who have relatively lower 2-back activation than would be expected given their higher activation in the 0-back condition, show the lowest EEA.

In the HCP sample, the SMN showed a pattern that was directly opposed to that shown by the task-positive networks; individuals in the lower right quadrant, who had relatively lower 2-back activation than would be expected given their higher 0-back activation, showed the highest EEA, while individuals in the upper left quadrant, with relatively greater 2-back

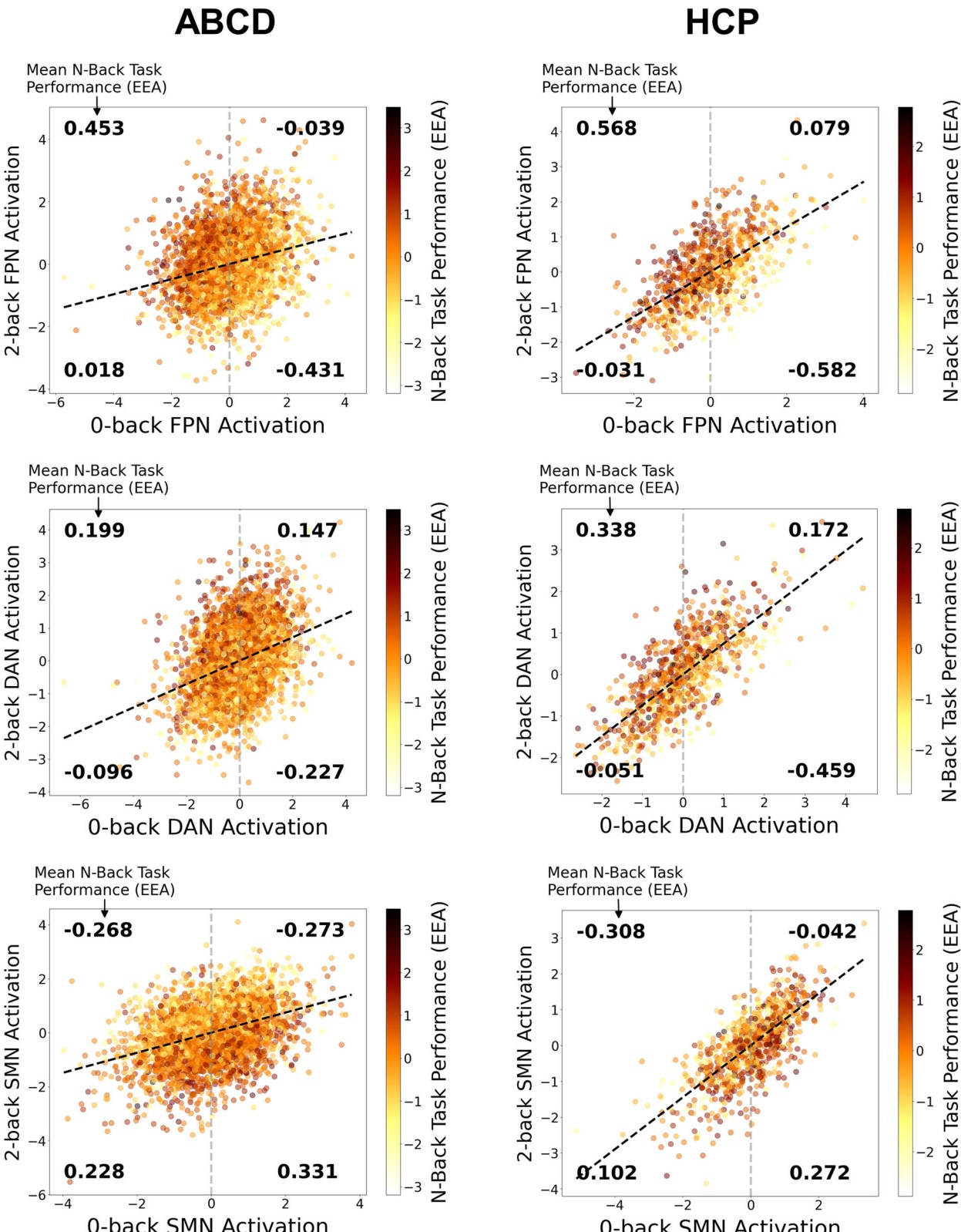

**Fig. 3 | Visualization of dynamic relations between average brain network activation in the 0-back and 2-back conditions and overall efficiency of evidence accumulation (EEA) on the task for the Adolescent Brain Cognitive Development Study (ABCD) and Human Connectome Project (HCP) samples.** All values were adjusted for age, sex, race/ethnicity, and motion covariates and were then converted to standardized scores (Z-scores: mean = 0, SD = 1) for interpretability. Activations of the frontoparietal network (FPN), dorsal attention network (DAN), and somatomotor network (SMN) across both levels of load are represented as scatterplots. Individuals' EEA is represented by the hue of the points, with individuals higher in EEA having darker red hues. Black dotted lines represent the regression line for relations between 0-back and 2-back task activations. Combined with the gray dotted lines representing the average 0-back activation level, the regression lines form four quadrants that denote whether individuals have higher or lower 2-back activation than would be expected given their level of 0-back activation. Bold numbers reflect the average EEA of individuals in each quadrant.

activation than 0-back activation, showed the lowest EEA. In contrast, in the ABCD sample, EEA seemed to be most clearly related to 2-back activation independent of 0-back activation, given that the two quadrants with 2-back activations below the regression line showed systematically higher EEA than the two quadrants with 2-back activations above the regression line.

These results suggest that, because of the strong positive dependency between activations from different levels of cognitive demand, the absolute level of FPN and DAN activation during a given task condition is less important for predicting EEA than the relative difference in activation between high-demand and low-demand conditions. Individuals for whom this difference is large simultaneously exhibit relatively higher activation in the high-demand (2-back) condition and relatively lower activation in the low-demand (0-back) condition, which appears to be a key signature of better EEA in task-positive networks. Consistent with this idea, the 95% CIs in Table 1 indicate that the absolute values of FPN and DAN 2-0 difference scores' associations with EEA were clearly systematically stronger (as indicated by non-overlapping CIs) than the associations of 0-back and 2-back activations alone with EEA, with one exception. In ABCD, the DAN 2-0 difference association ($r = 0.21$, CI = 0.19–0.24) was numerically stronger than the association involving 2-back activation alone ($r = 0.17$, CI = 0.14–0.21), but there was substantial overlap between the 95% CIs. When we estimated the 95% CI for the absolute difference between the DAN 2-0 and DAN 2-back alone $r$-values in ABCD, we found that it overlapped slightly with 0 (CI = 0.00–0.08). Therefore, there is strong evidence that 2-0 differences in FPN activation have more relevance to EEA than single-condition estimates and somewhat weaker evidence that the same is also true for 2-0 differences in DAN activation.

In HCP, the association of SMN 2-0 activation differences with EEA similarly had an absolute value that was systematically stronger than the associations of single-condition activations (non-overlapping CIs in Table 1). However, in ABCD, the CI for the SMN 2-0 difference association ($r = -0.30$, CI = -0.27– -0.32) was almost entirely overlapping with that of the association involving 2-back activation alone ($r = -0.29$, CI = -0.26– -0.32) and the CI for the difference between these $r$-values clearly overlapped with 0 (CI = -0.04– 0.03). Therefore, in contrast to the pattern in HCP, associations between SMN activation and EEA in ABCD appear to be driven by 2-back activation alone rather than by 2-0 activation differences.

### EEA is associated with flexibility in the engagement of the task-positive networks across different levels of cognitive demand

The findings detailed above suggest that individuals with higher EEA on the n-back tend to be those whose task-positive networks show higher activity in high-demand conditions but lower activity in low-demand conditions. To directly investigate this possibility, we sorted individuals in each sample into 10 bins ordered by EEA (adjusted for all covariates) and plotted each bin's mean FPN and DAN activations (Fig. 4). We also display regression lines for the association between bin order and activation to illustrate the overall pattern. Across both task-positive networks and both samples, greater EEA was associated with increases in network activity during the 2-back and corresponding decreases in network activity during the 0-back. Put another way, individuals with the lowest EEA engage these networks to a similar degree regardless of cognitive demands. In contrast, individuals with the highest EEA display the greatest degree of flexible modulation of task-positive network engagement in response to demands, exhibiting *both* the highest activation in task-positive networks in the 2-back and the lowest activation in the 0-back.

The SMN displayed a similar pattern of increased flexibility with greater EEA, but the pattern is the reverse of that of task-positive networks. The absolute values of the activation estimates in Fig. 4 indicate that the SMN is generally deactivated during task performance and is deactivated to a greater degree during the task condition with greater cognitive demands (2-back). Individuals with higher EEA display greater differences in deactivation between the 0-back and 2-back task conditions than individuals with low EEA. Although this effect appears be driven by both 0-back and 2-back activations in HCP, it appears to be primarily driven by 2-back

activation in ABCD, consistent with the finding reported in the previous section that associations between SMN activation and EEA in ABCD are driven by 2-back activation rather than by 2-0 differences.

### Differences between samples suggest potential developmental changes in task-positive network engagement

Although most brain network activation patterns and their associations with EEA were remarkably consistent across children in the ABCD sample and adults in the HCP sample, several key differences also emerged. As noted above, adults in HCP showed larger activation effects in the 2-0 contrast (despite showing similar spatial patterns of activation) than children in the ABCD sample. Adults also exhibited a much stronger positive dependency in task-positive network activations across high- and low-load conditions compared to children. In addition, the values displayed in Fig. 4 suggest possible developmental differences in overall levels of DAN activation under low cognitive demands. DAN activations in the 2-back were generally positive on average for both samples but, in the 0-back condition, ABCD participants' DAN activations were negligible or negative on average whereas HCP participants' DAN activations were consistently positive on average. Methodological differences in the task design (e.g., the use of face images from different emotion categories in ABCD) and the sampling strategies used in the ABCD and HCP studies preclude strong conclusions about developmental change. However, this pattern of results provides preliminary evidence that children do not consistently engage the DAN in low-load conditions and that improvements in performance in adulthood could be partially attributable to greater DAN engagement.

### Discussion

A growing literature on computational evidence accumulation models suggests that EEA, the rate at which a person gathers goal-relevant evidence to make adaptive choices, is a foundational mechanism that drives individual differences across many cognitive functions and has clear relevance to psychiatric disorders[5,6,19]. Although evidence accumulation processes are well-characterized at the level of discrete neurophysiological recordings during decision making[2,20–22,25,57,58], the role of largescale brain networks in supporting individual differences in EEA in humans remains poorly understood. The current study is the first to document a pattern of largescale brain network dynamics that shows strong and generalizable associations with individual differences in EEA across two large, diverse samples of children and adults. We demonstrate that neural responses to cognitive demand on the n-back can account for a large portion of the variance in EEA on the task. This association is largely driven by neural responses across the FPN and DAN, two "task-positive" networks involved in the control of attention and goal-directed cognition, as well as by patterns of task-related deactivation within the SMN, a network of regions involved in sensory processes and motor coordination. Crucially, we find a divergent pattern in which individuals with higher EEA exhibit *both* higher activity in task-positive networks during a difficult task condition with high cognitive demands (2-back) as well as lower activity in these same networks during a less demanding task condition (0-back). Although these findings are consistent with prior work suggesting that evidence accumulation processes are supported by FPN regions[36,40], the current study goes beyond this work in demonstrating a critical role for flexible adaptation of task-positive networks. That is, we demonstrate that dynamic changes in the activity of task-positive networks across different levels of cognitive demand, rather than these networks' static properties, are closely linked to EEA.

This set of findings naturally raises the question of how these complex network dynamics relate to the process interpretation of EEA in the cognitive modeling framework. As EEA is a formal index of the extent to which an individual can selectively parse goal-relevant evidence from noise in order to make adaptive choices during behavioral tasks[19], its opposing relations with task-positive network activity across the 0-back and 2-back conditions could reflect the modulation of attention. Specifically, during a difficult task that requires significant attentional resources (2-back), individuals with high EEA may allocate more attention to task-relevant features,

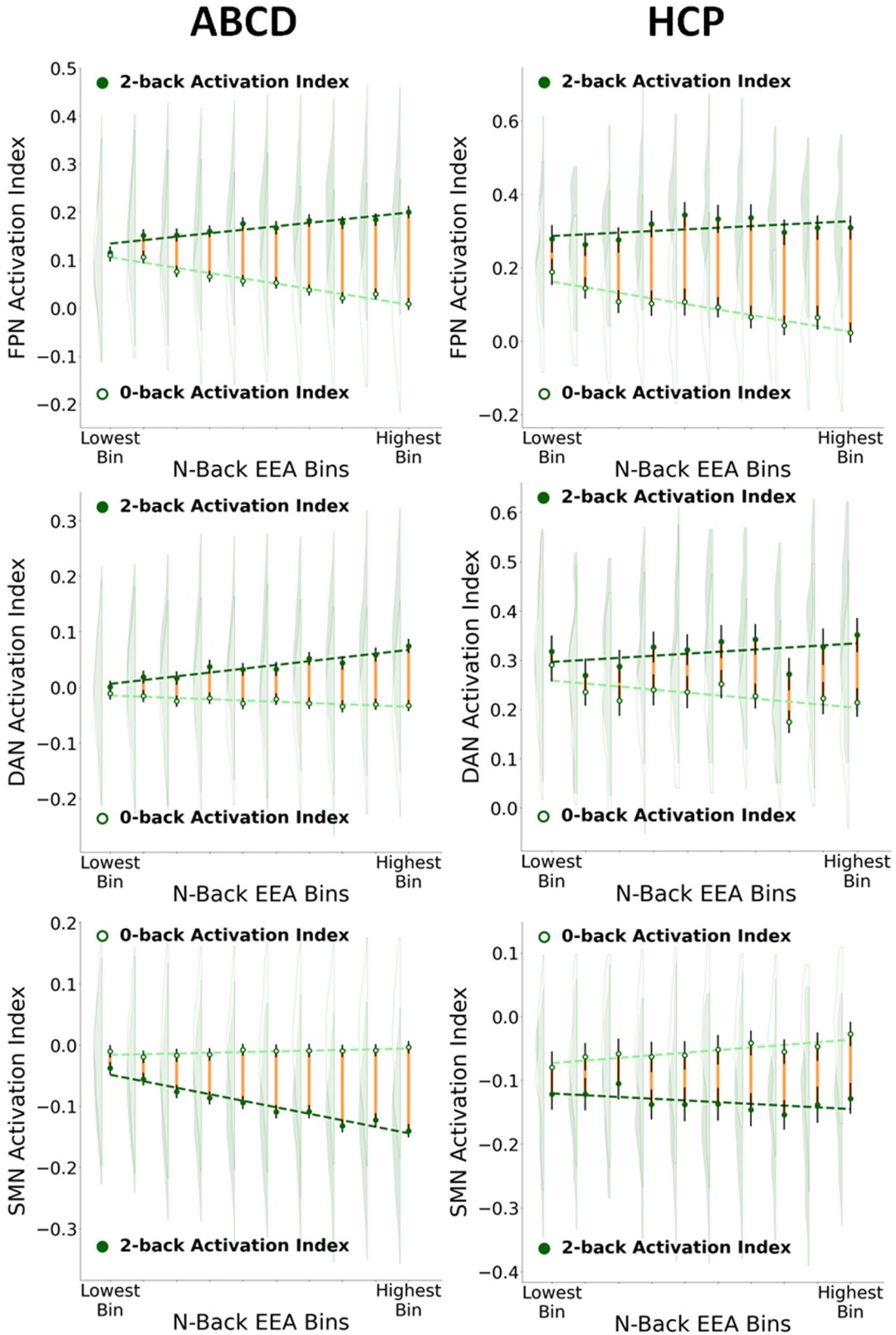

**Fig. 4 | Plots of mean frontoparietal network (FPN), dorsal attention network (DAN), and somatomotor network (SMN) activations ("activation index" = mean of activations across all vertices in the network) in the 0-back and 2-back conditions for 10 bins of individuals ranked by their mean EEA (covariate adjusted) in the Adolescent Brain Cognitive Development Study (ABCD) and Human Connectome Project (HCP) samples.** Vertical green lines around each point represent 95% confidence intervals while yellow lines highlight the difference between 0-back and 2-back means. Dashed lines represent regression lines for the relation between bin rank and each network activation index. Shaded and unshaded density plots represent the distribution of individuals' activation values for the 2-back and 0-back conditions, respectively (extreme values outside the 0.025 and 0.975 quantiles of each distribution were removed from the density plots to improve visualization).

resulting in greater 2-back activation in networks linked to attentional control and external task engagement. However, when the task at hand is relatively easy and does not require significant attentional resources to perform, individuals with high EEA may re-allocate attention away from this task in order to preserve attentional resources, leading to decreased activation in the same networks. These complementary processes, the allocation of attention to tasks that strongly require it and the efficient redistribution of attention during those do not, both arguably reflect flexible modulation of attention in response to environmental demands.

Such an explanation is notably consistent with the established conceptualization of attentional modulation posited in "adaptive gain" theory[53]. This theory posits that the norepinephrine system supports cognitive performance by optimizing an individuals' trade-off between exploitation of specific tasks and exploration of the larger environment. When task performance is not optimal, adaptive gain theory posits that the norepinephrine system works in coordination with prefrontal cortical areas to increase neural signal-to-noise ratios through the allocation of attention to the task-relevant stimuli. However, when further allocation of attention to a task is less likely to be rewarding for an individual (e.g., on a less demanding task), the same systems serve to disengage attention from the task at hand in order to re-allocate cognitive resources to the exploration of the environment in search of more rewarding behaviors. As EEA in cognitive models is a formal representation of the signal-to-noise ratio in the decision process, adaptive gain theory has been invoked to explain findings of poorer EEA in ADHD as reflecting neural systems' failure to flexibly modulate attention and arousal in response to external task demands[48,49]. The pattern of largescale brain network activation linked to EEA in the current study is consistent with this explanation, as it indicates that individuals with lower EEA engage networks involved in control of attention and external task processing to a similar degree regardless of task difficulty, whereas those with higher EEA flexibly modulate the engagement of these networks in response to changes in difficulty. Therefore, future work guided by adaptive gain theory may show promise for linking these network dynamics to the functioning of the norepinephrine system, especially given recent experimental findings indicating that norepinephrine agonists enhance individuals' EEA[14,31,59].

Our findings also bear interesting connections with previous theories regarding neural correlates of general cognitive ability. A large literature finds that individual differences in general cognitive ability are linked to conflicting patterns of neural recruitment across several neuroimaging modalities, a pattern of findings that was reconciled in a review proposing that these effects are moderated by cognitive demand[60]. More specifically, individuals with higher general cognitive ability show less recruitment of neural resources in lower-demand tasks (greater "neural efficiency") but also recruit more resources in tasks with high demands[60]. As EEA appears to be a key driver of general cognitive ability[5,6], the opposing relations between network activations at different levels of cognitive demand and EEA that are reported in the current study are broadly consistent with this theory. Importantly, this theory differs somewhat from the "adaptive gain" explanation detailed above; it posits that individuals with better cognitive performance differ in two distinct processes—greater neural efficiency on easy tasks and a separate tendency to allocate more neural resources to difficult tasks—rather than in the single process of flexible attention modulation. The current data cannot easily disambiguate between explanations, and future work would be necessary to do so. However, it should be noted that these explanations are not mutually exclusive and that the two-process explanation continues to imply greater flexibility in network modulation for individuals with higher EEA, as it posits that they require fewer neural resources to perform a given task well and that they are also able to flexibly allocate more of these resources when more are needed.

Our results also highlight another somewhat surprising phenomenon: despite showing strongly opposing relations with EEA, task-positive network activations in the high- and low-demand conditions are themselves positively correlated with one another, especially for adults in the HCP sample. These positive correlations, which were similarly observed in the SMN, may reflect systematic measurement errors that inflate individuals'

activation estimates across all conditions. Alternately, they may reflect a meaningful "general activation" dimension that represents individual differences in overall neural activation during most tasks, regardless of these tasks' level of cognitive demands. Regardless of the sources of these correlations, a key consequence of them is that activation in task-positive networks within single-task conditions appears to be more weakly related to EEA than differences in activation between high- and low-demand conditions. Indeed, across both samples, FPN activation measures drawn from only single levels of cognitive demand showed systematically weaker associations with EEA than measures of differences in FPN activation between high and low demands. The same pattern was observed for the DAN in HCP and a nonsignificant trend in the same direction was also observed for the DAN in ABCD. Therefore, from a practical standpoint of maximizing the prediction of EEA—and perhaps other cognitive variables—with task-related neuroimaging data, it follows that optimal prediction may require explicit experimental manipulations of cognitive demands in order to measure network adaptation, which the current study suggests is the most robust predictor of EEA.

Beyond the study's primary focus on the FPN and DAN, the results also have notable implications regarding two other largescale brain networks. Although unexpected, we found that neural activation in the SMN, a network comprised of both somatosensory cortices and primary motor cortex, played a role in predicting EEA that was comparably important to the roles of the FPN and DAN. However, the pattern of the SMN's predictive relationships was generally the reverse of the task-positive networks. The SMN showed a general pattern of task-related deactivation that was more pronounced in the high-demand (2-back) condition. Individuals who had higher EEA showed greater task-related SMN deactivations during the 2-back across both samples. In the HCP only, higher EEA was also associated with *reduced* task-related SMN deactivation in the 0-back condition. Taken together, these patterns provide preliminary evidence that EEA is not only related to flexible engagement of task-positive networks during demanding tasks, but may also be related to flexible disengagement of the SMN during the same tasks. The reasons for this pattern of disengagement are not clear and should be investigated in future work, including by attempting to identify whether specific areas of sensory or motor cortices show this pattern to a larger degree.

Another notable finding concerns the default mode network (DMN), a largescale brain network that is associated with off-task or internally-focused cognition and that also shows a canonical pattern of deactivation during the processing of external tasks[61,62], including the n-back[63,64]. Given the DMN's role in off-task processes, suppression of DMN activity has been hypothesized to be related to cognitive control and to facilitate successful cognitive performance[65,66]. Although the canonical pattern of reduced DMN activation during higher cognitive demand was observed in both the HCP and ABCD samples (Fig. 2a, b), we found that individual differences in these demand-related changes showed little evidence of any substantial relation to EEA in either sample (Table 1). This striking and unexpected null finding is inconsistent with prior hypotheses about the role of DMN suppression in cognitive performance. It is also relevant to prior work in the HCP that identified several discrete activation clusters in the DMN that were associated with 2-back task accuracy[67]. Taken together with this prior work, the current results suggest that n-back performance may relate to deactivation in discrete subregions of the DMN, but not deactivation measured at the level of the entire network.

Although we focused on canonical networks derived from resting state connectivity, it is worth pointing out that our results are relevant to the growing body of work on the "multiple demand network" (MDN), a set of structures that shows considerable overlap with the FPN and DAN and that appears to support cognitive performance across a wide array of tasks[68,69]. The generality of this network's relation to task behavior has led to the suggestion that the MDN is the basis of the common factor that explains covariance across many "executive function" tasks[56]. As EEA shows similar task-general properties and was recently demonstrated to be a strong explanation for the common factor of executive functioning[70], future work

on the MDN may be able to determine whether EEA mediates the relation between demand-related MDN dynamics and performance on diverse executive tasks. Our findings are also notably consistent with recent work demonstrating that children and adults tend to activate similar brain structures in the MDN during task performance but that children do so to a lesser degree[56].

This study has several limitations that should be considered and that may be addressed by future work. A primary limitation is that the study focuses exclusively on linking between-person differences in network dynamics to between-person differences in EEA. Additional evidence that these network dynamics are related to fluctuations in the evidence accumulation at the within-person level would provide stronger evidence that they support EEA. Investigating these within-person associations is made challenging by the fact that estimates of evidence accumulation and neural responses at the level of individual trials or blocks are likely to be very noisy given the reduced amount of data available. However, advances in hierarchical Bayesian joint modeling methods that estimate models of behavioral and neural data as being described by a single multivariate normal distribution[71–73] have recently shown great promise for allowing within-person links between brain and behavior to be characterized while minimizing measurement error. We believe that these methods can be used in the near future to better characterize both between- and within-person relations between task-positive network dynamics and EEA. Another limitation of the study with regard to understanding developmental differences is that the samples of child (ABCD) and adult (HCP) participants used in this study had different participant sampling strategies and slightly different n-back task designs, which precludes strong inferences about developmental differences. Future analyses of longitudinal data in the ABCD sample once participants mature would be helpful for better understanding the potential developmental differences identified in the current study. Finally, given recent indications that the length of response times can lead to systematic effects in fMRI activation estimates[74,75], it is worth considering the possibility that some of the network dynamics we identify are related to individual or condition-related differences in response times. We did not account for response times as a nuisance parameter in our model of task-related activation because the DDM assumes that evidence accumulation is directly related to the length of response times, and accounting for them in this way was therefore likely to remove neural signal of interest. It is currently unclear how accounting for response times should be handled in this situation, and future work should seek to explore how response times do or do not relate to the network dynamics we identify. However, we note that, as response times would be expected to be longer in individuals with lower EEA across both n-back load conditions, it seems unlikely that the opposing EEA associations of 2-back versus 0-back brain network activations can be explained by response times alone.

In conclusion, the current study characterizes dynamic properties of large-scale brain networks that show strong and replicable associations with EEA, a foundational cognitive individual dimension derived from a well-developed cognitive modeling literature. The findings specifically highlight individuals' ability to flexibly engage, versus disengage, the FPN and DAN in conditions of high, versus low, cognitive demand as a key underpinning of EEA. These findings suggest several productive future avenues of investigation into relations among largescale brain network dynamics, neurotransmitter systems thought to support flexible behavior, and EEA's downstream consequences for adaptive functioning and risk for clinical disorders.

## Methods

### HCP sample, task design, and data acquisition
Data for the HCP sample were taken from the HCP-1200 release[54,76]. Participants provided informed consent and all study procedures were approved by the Washington University IRB. All ethical regulations relevant to human research participants were followed.

Participants completed an n-back task in which they were presented with a series of images from different categories (faces, places, tools, and body parts) and were asked to respond in two conditions. In the 0-back condition, participants were shown a 'target' image during a 2.5 s cue at the beginning of each 10-trial block and were asked to respond as to whether each subsequent stimulus presented was the same as the cued target image. In the 2-back condition, participants were instructed by the 2.5 s cue at the beginning of the block to respond as to whether each stimulus presented in the block was the same image as the image presented 2 spaces back in the sequence. Participants completed 8 10-trial blocks in each condition across two neuroimaging runs for a total of 80 trials in each task condition. Each block contained images from the same image category and contained 2 target trials and 2–3 "lure" trials, recently presented stimuli that do not meet criteria for being targets. Each stimulus was presented for 2 s with a 500 ms interstimulus interval.

Participants completed two runs of the n-back task (~5 min each, TR = 720 ms, 2.0 mm isotropic voxels). High resolution (0.7 mm isotropic) T1-weighted and T2-weighted images were also collected and used for data processing. Comprehensive details are available elsewhere on HCP's overall neuroimaging approach[54,77] and HCP's task fMRI dataset[78].

Following exclusions for neuroimaging and behavioral data quality (described below), a total of 883 participants (465 females; mean age = 28.6, SD age = 3.7) were included in analyses.

### ABCD sample, task design, and data acquisition
The ABCD Study® is a multisite longitudinal study with 11,875 children between 9 and 10 years of age from 22 sites across the United States[55]. ABCD study procedures were approved by the University of California San Diego's IRB, which serves as a single IRB for the ABCD consortium, as well as by the corresponding IRBs of each participating study site. All participants provide informed consent (parents) or assent (children). All ethical regulations relevant to human research participants were followed. Data for this study are from ABCD Release 4.0.

Participants completed an "emotional" n-back task[55] that had a design identical to the HCP n-back task with regard to the experimental conditions (0-back, 2-back), block and trial structure, event timing, and number of trials. The primary difference between this task and the HCP n-back task is the image categories that are used. Rather than images of faces, places, tools, and body parts being presented in a given block, the ABCD "emotional" n-back variant contains blocks that present faces with happy, fearful or neutral facial expressions as well as blocks that present images of places.

High spatial (2.4 mm isotropic) and temporal resolution (TR = 800 ms) resting state fMRI was acquired for the emotional N-back task in two separate runs (~5 min each). High resolution (1 mm isotropic) T1-weighted and T2-weighted images were also collected and used for data processing.

Following exclusions for neuroimaging and behavioral data quality (described below), a total of 4315 participants (2182 females; mean age = 10.0, SD age = 0.63) were included in analyses.

### Neuroimaging data processing
Preprocessing was performed using fMRIPrep version 1.5.0[79]. Briefly, T1-weighted (T1w) and T2-weighted images were run through recon-all using FreeSurfer v6.0.1. Functional data were corrected for fieldmap distortions, rigidly coregistered to the T1, motion corrected, normalized to standard space, and transformed to CIFTI space with 91,282 grayordinates. All preprocessed data for each individual run then went through two stages of visual inspection procedures that were carried out by one of the authors as well as by a team of trained undergraduate research assistants under their supervision. First, the mean functional image for the run was overlaid on the anatomical (T1) image for the subject to check for problems with co-registration between the functional and structural images. Runs for which the functional and structural images displayed clear lack of overlap were removed from further analysis. Then, the MNI space mean functional image was overlaid on the MNI space T1 for the subject to check for any issues with the warping to MNI space. Runs in which the warped images displayed gross abnormalities were removed from further analysis.

Task models were constructed following HCP scripts. Models were constructed using FSL (6.0.5.2) in a two-stage procedure, estimating each run, and then averaging runs. CIFTI data were smoothed with a 2 mm FWHM Gaussian using HCP Connectome Workbench (1.4.2). Images were then high pass filtered at 0.005 Hz. Nuisance covariates in the first level models consisted of 24 motion correction parameters (3 rotation, 3 translation, first derivatives of each, and quadratics of original and derivatives), top 5 principal components of signal from white matter, top 5 principal components of signal from cerebrospinal fluid, and individual regressors for each TR that exceeded a 0.9 mm framewise displacement. Task conditions, which included separate conditions for 0-back and 2-back blocks and for each category of stimuli (HCP: faces, places, body parts, tools; ABCD: happy faces, neutral faces, fearful faces, places), were all modeled with separate regressors (e.g., 0-back places, 2-back places, 0-back tools, 2-back tools, etc.). The baseline was left unmodeled, resulting in an implicit baseline. Linear contrasts were constructed for the 0-back, 2-back, 2-back minus 0-back conditions of interest. Each contrast was made to average over all of the relevant 0-back and/or 2-back conditions.

Individual runs were considered good if they passed visual inspection and had at least 4 minutes of uncensored data. Subjects were only included if they had two good runs and complete task behavioral data that met the inclusion criteria described in the section below.

## EEA estimation
EEA was estimated by fitting the diffusion decision model (DDM), a widely used evidence accumulation model[80], to data from the n-back task in both the ABCD and HCP samples using Bayesian estimation methods implemented within the Dynamic Models of Choice (DMC)[81] suite of R functions. In both samples, participants completed 80 trials in each of the two cognitive load conditions (0-back, 2-back). Specific details of the stimuli and task parameters are described in detail elsewhere[54,55,78]. At both levels of load, trials could be (1) "target" stimuli, which meet specific criteria for being targets (e.g., in the 2-back, stimuli that were previously presented exactly 2 spaces back), (2) "novel" stimuli, which are stimuli that have never been presented before, (3) "lure" stimuli, which were recently presented stimuli that do not meet the specific criteria for being targets. Lures are more difficult for participants to reject and ensure that participants are applying the full target criteria while completing the task rather than relying on the familiarity of stimuli, alone. The DDM included eight parameters for each level of cognitive load (0-back/2-back): three separate drift rate ($v$) parameters for target, novel, and lure stimuli, single boundary separation ($a$), non-decision time ($t0$), non-decision time variability ($st0$), and start point ($z$) parameters, and a parameter for the probability of "contaminant omissions", which are non-responses due to causes outside of the main DDM response process (e.g., inattention)[82]. Omissions due to the task design (i.e., response cut off by the 2-s response window) were also addressed using methods developed in prior work on addressing omissions with evidence accumulation models[82]. Parameters for between-trial variability in the drift rate ($sv$) and starting point ($sz$) were not estimated due to difficulties with accurately recovering these parameters without large numbers of trials[83].

Prior to estimation, we excluded individuals' data if they displayed accuracy rates close to chance (<55%) or excessive rates of omissions/non-responses (>25%) in a given load condition, both of which indicate likely disengagement from the task. We also excluded RTs <200 ms as these RTs are likely to reflect fast guesses by participants. Informative priors for parameter estimates were generated following a procedure we previously developed[84]. A hierarchical version of the DDM was fit to an independent sample of 300 ABCD participants who had failed neuroimaging data quality checks but not behavioral data quality checks and who were unrelated to the ABCD participants included in the main analyses of this study. Following parameter estimation for this independent subsample, we fit truncated normal distributions to the full distribution of all individual-level posterior samples for each parameter. These truncated normal distributions were then used as informative priors for model fits in the ABCD sample. For the

HCP sample, we multiplied the scale of these priors by 1.5 to make them slightly less informative given that the adults in HCP likely display some developmental differences relative to the prior-generation sample drawn from children in ABCD. Sensitivity analyses (Supplemental Materials) suggested that our choices of priors had a negligible impact on inferences drawn from the study.

The DDM was then estimated under informative priors at the individual level using the automated RUN.dmc() function that repeats the posterior sampling process until convergence is obtained (rhat <1.1). Convergence was corroborated by visually inspecting a subset of individuals' sampling chains. Model fit was assessed using posterior predictive plots[85], which indicated that the model provided an adequate description of the behavioral data in both samples (Supplemental Figs. 1 and 2). Posterior medians for the drift rate ($v$) parameter were averaged across the three types of stimuli (target, novel, lure) to index individuals' EEA at each level of cognitive load.

## Multivariate predictive modeling
Individuals' vertex-level activation data from across the entire cortical surface for the 2-back minus 0-back contrast were used in a cross-validated principal components regression (PCR) predictive model[42]. In brief, this method performs dimensionality reduction on the input data (in this case, individuals' 2-0 contrast activation at each spatial location across the entire cortex), fits a regression model on the resulting components, and applies this model out of sample in a 10-fold (HCP) or leave-one-site-out (ABCD) cross-validation framework. Nuisance covariates (age, age squared, sex, race/ethnicity, framewise displacement estimate of motion, framewise displacement squared) are handled by calculating a cross-validated form of partial correlation. After a principal components analysis (PCA) is conducted to reduce the data in each training fold, K components are retained, with the optimal value for K being estimated with a nested 5-fold cross-validation within the training data only. Both the component expressions as well as the outcome variable are then regressed against nuisance variables. The betas estimated from this model are used to residualize both the training and test data and then a linear model is fit on the training data to predict the residualized outcome with the residualized expressions. This model is then applied to the test data to obtain a predicted value, which can then be correlated with the residualized outcome in the test data to obtain an out-of-sample partial correlation estimate. This is repeated for each fold and the per-fold correlations are then averaged across folds. Consensus maps that indicate the relative importance of 2-0 activation from each cortical surface area for predicting EEA were generated by multiplying the component loadings of each vertex by the betas for each component in the predictive model of EEA and summing these values across all components for that vertex. The resulting values for each vertex were then displayed on the cortical surface using a color scale. We include a detailed visualization of the steps involved in the multivariate predictive modeling process in Supplemental Fig. 3.

## Network activation extractions
Average activation values for each of the networks in the 7-network Yeo[86] parcellation were estimated by taking the mean across all vertices within each network for each contrast of interest. These average activation estimates were computed separately for the 0-back and 2-back contrasts (relative to implicit baseline) as well as for the cognitive load contrast (2-back minus 0-back).

## Relations among network metrics and EEA
Following predictive modeling analyses and extractions of network activation means for the FPN, DAN, and the five other networks in the Yeo[86] parcellation, we assessed correlation coefficients ($r$) for relations among EEA and 0-back, 2-back, and cognitive load contrast (2-back minus 0-back) average activation values for each network. For all analyses involving EEA and these network activation averages, nuisance covariates (age, age squared, sex, race/ethnicity, framewise displacement estimate of motion,

framewise displacement squared) were addressed by fitting multiple regression models in which each variable of interest was predicted by all nuisance covariates. Residuals from these models were used in all analyses and plots except for the binned EEA plots in Fig. 4, which used raw values of all variables for interpretability. Sensitivity analyses revealed no substantive differences between inferences drawn from raw versus covariate-residualized values (Supplemental Materials; Supplemental Table 2; Supplemental Fig. 4). A clustered bootstrapping procedure was used to estimate 95% confidence intervals (CIs) for *r* while accounting for nesting of individuals within families and ABCD sites.

## Statistics and reproducibility
Multivariate predictive modeling was conducted within Python and statistical analyses of summary network activation metrics were conducted within R. Python and R code for all analyses can be accessed at: https://osf.io/yte76/. All analysis procedures were replicated exactly across the ABCD and HCP data sets. As described above, we used 95% confidence intervals estimated via clustered bootstrapping for statistical inference.

## Reporting summary
Further information on research design is available in the Nature Portfolio Reporting Summary linked to this article.

## Data availability
The ABCD data used in this report came from ABCD release 4.0 (https://nda.nih.gov; https://doi.org/10.15154/1,523,041). ABCD data specific to the current report can be accessed in NDA Study 2297 (https://doi.org/10.15154/wnt8-dq37). HCP data are accessible at: https://db.humanconnectome.org/. Source data for Figs. 1, 3 and 4 are available in Supplemental Data 1. Source data for Fig. 2 are available at: https://figshare.com/s/ab78c31c258e5c3e36b3.

## Code availability
Code for all study analyses can be accessed at: https://osf.io/yte76/.

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

## Acknowledgements

A.W. was supported by K23 DA051561 and R21 MH130939. C.S. was supported by the Dana Foundation David Mahoney Neuroimaging Program. A.W., A.H., and C.S. were supported by R01 MH130348. C.S. and M.H. were supported by U01 DA041106. A.H. was supported by Vidi grant VI.Vidi.191.091 from the Dutch Research Council. Data used in the preparation of this article were obtained from the Adolescent Brain Cognitive Development^SM (ABCD) Study (https://abcdstudy.org), held in the NIMH Data Archive (NDA). This is a multisite, longitudinal study designed to recruit more than 10,000 children aged 9–10 and follow them over 10 years into early adulthood. The ABCD Study® is supported by the National Institutes of Health and additional federal partners under award numbers U01DA041048, U01DA050989, U01DA051016, U01DA041022, U01DA051018, U01DA051037, U01DA050987, U01DA041174, U01DA041106, U01DA041117, U01DA041028, U01DA041134, U01DA050988, U01DA051039, U01DA041156, U01DA041025, U01DA041120, U01DA051038, U01DA041148, U01DA041093, U01DA041089, U24DA041123, U24DA041147. A full list of supporters is available at https://abcdstudy.org/federal-partners.html. A listing of participating sites and a complete listing of the study investigators can be found at https://abcdstudy.org/consortium_members/. ABCD consortium investigators designed and implemented the study and/or provided data but did not necessarily participate in analysis or writing of this report. This manuscript reflects the views of the authors and may not reflect the opinions or views of the NIH or ABCD consortium investigators. The ABCD data repository grows and changes over time. The ABCD data used in this report came from ABCD release 4.0 (https://nda.nih.gov; https://doi.org/10.15154/1,523,041).

## Author contributions

Conceptualization: A.W., C.S., A.T., M.A., A.H., and M.H; methodology: A.W., C.S., A.T., M.A., and A.H.; formal analysis: A.W., A.T., and M.A.; data curation: M.A. and A.T.; writing—original draft: A.W.; writing—reviewing and editing: A.W., C.S., A.T., M.A., A.H., and M.H.; visualization: A.T. and M.A.; funding acquisition: M.H., C.S., and A.W.

## Competing interests

The authors declare no competing interests.
