## [Peer Review File · Communications Biology]

Reviewers' comments:

Reviewer #2 (Remarks to the Author):

Reviewer's comments on the manuscript:

Flexible adaptation of task-positive brain networks predicts efficiency of evidence accumulation

Summary:

In the present manuscript, the authors investigated the neural basis of EEA on the Human Connectome Project dataset fitting a diffusion decision model for EEA estimation and performing multivariate predictive modeling. The authors hypothesized that the flexible adaptation that can be linked to the activity in FPN and DAN supports efficient cognitive processing and EEA. They tested it on two datasets that cover different developmental periods and predicted the neural responses to cognitive demand explain a great portion of the variance in individuals' EEA. They were able to verify their previously introduced hypothesis about a strong association between EEA and flexible adaptations to cognitive demands. The manuscript is of good quality, although more explanations on the methods are needed to fully understand what analyses the authors performed and to follow the rationale behind the methods selection and the rationale behind them. Moreover, the discussion could be more diverse.

Comments:

1. Abstract: it is mentioned that data from over 5000 participants was used, better to mention the exact number
2. Introduction (page 2, last paragraph): Work in non-human primates: if in the scope, perhaps there could be more information provided on this. What kind of properties are meant here? On which data, which kind of experiments?
3. Introduction: Here, only DAN and FPN are discussed, would EEA also be associated with e.g. CCN (cognitive control network)? Are there any findings about negative associations, e.g. with DMN?
4. Methods: it is stated that the HCP dataset subjects completed the N-back working memory task, whereas the ABCD subjects underwent an emotional N-back task? What is the difference? Are these comparable? Was it somehow controlled for it taking other brain networks associated with emotional processing or empathy or similar into account, e.g. by contrasting activations?
5. Methods: The number of ABCD participants is way higher, is this not driving the results?
6. Methods: page 15: it is mentioned that the individual runs underwent visual inspection. What were the criteria in this case? Which runs passed the visual inspection?
7. Methods: EEA estimation: What are the two other variability parameters that could not be included in the estimation?
8. Methods: "For the HCP sample, we multiplied the scale of these priors by 1.5 to make them slightly less informative given that the adults in HCP likely display some developmental differences relative to the prior-generation sample drawn from children in ABCD". Why multiply by 1.5? Is there some theoretical background for this number or is this arbitrary?
9. Methods: Multivariate Predictive Modeling: having a visualization of this process would be very

helpful, even in the supp materials. This is otherwise very difficult to follow and to grasp what the authors did in the analysis.

10. Results: “We tested their generalizability in independent data using leave-one-site-out cross-validation in ABCD and 10-fold cross-validation in HCP” – why did the authors not use the same here? The rationale behind this is unclear.

11. Results: Figure 1 B: this figure is contrainuitive. Could the authors please add a dotted line as the average on the right plot, as is done on the left plot? Could these two plots be more similar so that the message behind the figure is clear?

12. Results: 6: could this be driven by multiplying the weights by 1.5? What is the outcome of the analyses if you keep the weights as they are or change the multiplication factor?

13. Discussion: Perhaps the authors could discuss some limitations of their approach. Moreover, the discussion could be shedding light on more perspectives of the topic.

Reviewer #3 (Remarks to the Author):

I wish to thank the editors and authors for the opportunity to review this interesting manuscript. In this important work, the authors explore the neural underpinnings of Efficiency of Evidence Accumulation (EEA) in an N-back task using two large HCP samples. Their findings reveal that activity within two large-scale networks of an influential 7-network whole-brain parcellation – the Frontoparietal (FPN) and Dorsal Attention Networks (DAN) – is associated with EEA. Furthermore, the ability of these networks to track task demands, increasing when the demand is greater and decreasing during easier conditions, seems to explain more variance than any one condition by itself, hinting at the importance of flexible reconfiguration of neural resources in these networks as an underpinning of this cognitive process, important for general intelligence. This is impactful work that sheds light over a general mechanism of cognitive performance and is thus likely to be of interest to a wide range of cognitive neuroscientists. I would recommend publication provided the authors can address a few concerns.

MAJOR

1. On paragraph #1 of the discussion: The authors state that “We demonstrate that neural responses to cognitive demand on the n-back can account for a large portion of the variance in EEA on the task and that this association is largely driven by neural responses across the FPN and DAN”. The second part of this statement is difficult to sustain on the evidence presented. They did not compare, to the best of my understanding, the FPN and DAN with the five remaining Yeo networks. Thus, the association could be driven by one of the unexamined networks, but their statement seems to imply it is these two networks in particular that best explain the variance. If they wish to make this statement, I believe they should be comparing the variance explained by these two networks vs the rest of the parcellation.

2. In the same paragraph, the authors state “Indeed, FPN and DAN activation measures drawn from only single levels of cognitive demand showed systematically weaker associations with EEA than measures of differences in FPN and DAN activation across high- and low-demand levels.”. However, to the best of my understanding they did not perform a direct comparison of strength of association

between one vs multiple conditions, nor of the amount of variance explained to sustain this statement. Near the middle of page 9 the authors mention that variance explained is higher in the between-conditions contrast relative to within-conditions, but the % numbers presented within brackets are quite close to each other and we do not know whether these were statistically significantly different. It is difficult to know if a difference of 2-5% points is relevant. Carrying out formal statistical testing would allow the reader to reach their own conclusion. As a formatting issue, it would be better to present the percentages of variance explained in Table 1.

3. Point #4 on pp 7-9 seems mostly descriptive. The authors show the striking pattern of people with low activation on the 0-back and high activation on the 2-back condition having better performance on the task (or EEA), but they do not indicate how they assessed statistically whether this relationship is significant at the individual level. In point #5 they perform an analysis binning the participants in 10 bins sorted by EEA and demonstrate that, at this 'group' level the relationship seems to hold (as an aside, the authors should clarify what is the 'brain activation index?'). Their case would be strengthened if they could show this relationship is significant at the individual level, though. Are they able to identify these relationships in individual contrasts? Is a subtraction of activation in 2-0 positively (and significantly) associated with EEA for the top performers, but negatively with the bottom performers? Can you test difference scores, for example? Or incorporate EEA as a covariate? I believe carrying out analyses that demonstrate the existence of this relationship at the individual level would constitute irrefutable evidence for the conclusions this paper draws.

4. On page 7, point #3, the authors mention a 'Gordon parcellation'. It hasn't been mentioned before, and the rest of the text implies the authors used Yeo's parcellation. Is this a typo? Which parcellation was used? Can they elaborate on and add a reference for the Gordon parcellation if it's not?

MINOR

The authors could substantially improve the manuscript by elaborating on some of the methodology as outlined in the points below:

- P14, The authors should be more explicit regarding how they modelled the N-back tasks. Did they have an implicit baseline? If so, what was it? Were the linear contrasts (e.g., 2-back vs 0-back) collapsing across all task conditions, or was it one for each condition which were later averaged?
- P15, when the authors state "Individual runs were considered good if they passed visual inspection", can they elaborate on what exactly they were checking for during visual inspection? I find it impressive that they visually inspected 5k+ subjects and would like to hear more about how thoroughly and under which criteria these data were inspected.
- P15, EEA Estimation: Even though the tasks have been described in detail elsewhere, it would be good if the authors included a brief description of the tasks/conditions used for ease of reading.
- P16, Multivariate Predictive Modelling: What variables were submitted to dimensionality reduction in the PCR to obtain the components? Was it 3-dimensional contrast images (i.e., was the actual data the voxels of the contrast results per subject)? A more elaborate, detailed description would be good for subjects not acquainted with the method.
- P16, Network Activation Extractions: What do the authors mean with "the contrasts averaged within each network"? Did they average the activation of the 0-back and 2-back contrasts? What is the logic of this? Or do they mean they averaged the activation within each network of the Yeo7

parcellation?

- P16, Relations Among Network Metrics and EEA: What variables were actually correlated? Was it the v parameters of the DDM with mean activation value per network (for each condition/contrast)? How did the authors deal with the different task categories when producing these contrasts? This seems important to outline explicitly.
- The authors could elaborate on how consensus maps work for readers that might be unfamiliar. What are they? How were they constructed? This information could be added to the methods.

We wish to thank the reviewers and editor for their careful consideration of our manuscript and for the helpful and constructive feedback we received. We have now taken the opportunity to revise the manuscript in response to reviewers' specific comments, including the addition of several new analyses that we believe have made the paper more comprehensive and impactful. For clarity, reviewers' comments are shown in *italics*, and our responses are shown in indented text.

Reviewer #2

The manuscript is of good quality, although more explanations on the methods are needed to fully understand what analyses the authors performed and to follow the rationale behind the methods selection and the rationale behind them. Moreover, the discussion could be more diverse.

We thank the reviewer for their kind words about our work and the constructive comments that follow.

1. Abstract: it is mentioned that data from over 5000 participants was used, better to mention the exact number

We now report the exact number (5,198) in the abstract.

2. Introduction (page 2, last paragraph): Work in non-human primates: if in the scope, perhaps there could be more information provided on this. What kind of properties are meant here? On which data, which kind of experiments?

We have now added additional details about this line of work (pages 3-4), including the experimental oculomotor decision making tasks that are typically used in these studies, evidence for similarities between neural ramping patterns and the decision process assumed by evidence accumulation models, and recent work that provides stronger evidence for this link by simultaneously modeling behavioral and neural data.

3. Introduction: Here, only DAN and FPN are discussed, would EEA also be associated with e.g. CCN (cognitive control network)? Are there any findings about negative associations, e.g. with DMN?

Our study was primarily focused on the canonical brain networks identified in the 7-network Yeo parcellation because these networks have precise and standardized definitions that are used consistently across a wide array of studies. The term "cognitive control network" is often used less precisely to refer to regions involved in cognitive control (Witt et al., 2021; *Brain Topography*), but most studies we could find that use this term appear to be referring to a network that is largely or entirely overlapping with the canonical FPN (e.g., Rosen et al., 2016, *Cerebral Cortex*; Wu, Dufford, et al., 2018, *Cerebral Cortex*; Stang et al., 2017, *Human Brain Mapping*). Therefore, we would expect that the "cognitive control network", as generally defined in the field, would have an association with EEA that is highly similar to that of the FPN, as precisely defined in the standardized method used in this study.

The reviewer's point about the DMN is very pertinent, as we would certainly expect, based on prior findings that DMN activation is reduced under high cognitive load and is associated with inattentive performance, that greater *deactivation* of the DMN in high-

load tasks would predict EEA. To investigate this possibility, and to address similar concerns raised by Reviewer 3, below, we now assess relations between EEA and load-related change in neural activity EEA within all 7 canonical brain networks (Table 1). We were quite surprised to find that, although at least one network other than the FPN and DAN made substantial contributions to prediction of EEA, relations between DMN activity and EEA were non-significant in both the ABCD and HCP samples. We now discuss this unexpected and interesting null finding in detail on page 19 in the Discussion. We thank the reviewer for spurring us to look more closely at the DMN because we believe this finding is valuable to report to the field.

4. Methods: it is stated that the HCP dataset subjects completed the N-back working memory task, whereas the ABCD subjects underwent an emotional N-back task? What is the difference? Are these comparable? Was it somehow controlled for it taking other brain networks associated with emotional processing or empathy or similar into account, e.g. by contrasting activations?

We apologize for our lack of detail on the task design in the previous manuscript draft. As we now describe in greater detail in the Methods section (pages 22-23), the HCP and ABCD n-back task designs are identical with the exception that that image stimuli presented in HCP consisted of faces, places, tools and body parts whereas the image stimuli presented in ABCD consisted of faces with happy, fearful or neutral facial expressions as well as images of places. The design change in ABCD was intended to allow for researchers to investigate neural activations related to different emotional stimuli in studies focused on emotional processing.

As detailed on page 25, we controlled for image type within the first-level task models of both the HCP and ABCD tasks by having separate regressors for each image category. Although it is possible that the use of different images in each of the tasks may have led to activation differences in regions involved in emotion processing, we emphasize that the current study is focused on finding effects that generalize across both the HCP and ABCD samples rather than on differences between the samples. Hence, the fact that most of the associations we identify are replicated across both the HCP and ABCD samples despite the differences in stimuli used for the respective tasks underscores the robustness and generalizability of these associations. We now also highlight the caveat of differences in image stimuli in Results Section 6 (pages 13-14), in which we state that that any possible developmental effects that are inferred from comparisons between the two samples are necessarily preliminary because of differences in the task design and the sampling strategies used in each study. We also note this as a general limitation of the study in the Discussion on pages 20-21.

5. Methods: The number of ABCD participants is way higher, is this not driving the results?

We used two of the largest samples available for this imaging task and we analyzed each sample separately, so one sample cannot drive the effect in the other sample. Given this, the main consequence of one sample being larger than another would be that there would be more precise estimates of effects (i.e., tighter 95% CIs) and better statistical power to detect smaller effects in the larger of the two samples (ABCD). Despite the ABCD and HCP samples differing in size, both samples were large enough to reliably detect even small effects in inferential tests (85% power to detect $r=0.10$ effect in HCP; >99% power in ABCD). Furthermore, although the 95% CIs are, as expected, consistently wider in HCP due to the smaller sample size, the major findings of interest are highly similar across both samples. Given the robustness of the effects of interest

across samples and the study's goal of finding such generalizable effects, it is unlikely that the difference in sample size had an impact on the main findings of the study.

6. Methods: page 15: it is mentioned that the individual runs underwent visual inspection. What were the criteria in this case? Which runs passed the visual inspection?

We now report our standard visual inspection procedures in greater detail on page 24.

7. Methods: EEA estimation: What are the two other variability parameters that could not be included in the estimation?

We now clarify on page 26 that the two omitted parameters were for between-trial variability in the drift rate (sv) and starting point (sz). These parameters were omitted because they have been found to be difficult to estimate without large numbers of trials (Lerche, Voss, & Nagler, 2017, *BRM*).

8. Methods: "For the HCP sample, we multiplied the scale of these priors by 1.5 to make them slightly less informative given that the adults in HCP likely display some developmental differences relative to the prior-generation sample drawn from children in ABCD". Why multiply by 1.5? Is there some theoretical background for this number or is this arbitrary?

Setting priors in Bayesian analyses for which comparable independent data are not available necessarily involves some subjective considerations. We increased the scale by 50% because this seemed like a reasonable way to make the priors more weakly informative for the adult sample than they were for the child sample from which the original priors were derived. Ultimately, the best way to determine whether prior choice could influence substantive inferences is to conduct sensitivity analyses in which parameters that are estimated under different types of priors are compared. Therefore, we now report (page 27; Supplemental Materials, page 2) correlations between estimates of the main parameter of interest (EEA) that were estimated under the informative priors we used for each data set (the ABCD-derived priors for ABCD and the same priors with scales multiplied by 1.5 for HCP) and much broader and uninformative priors (Supplemental Table 1). In both samples, EEA estimates were almost perfectly correlated (ABCD $r = 0.97$; HCP $r = 0.99$) between the two different sets of priors. This is not necessarily surprising, as priors for the mean and scale of model parameters are likely to influence the absolute values of parameter estimates but are less likely to influence their rank ordering, which is the main focus of this study. Hence, we can be confident that prior choice has a negligible impact on inferences drawn from the study.

9. Methods: Multivariate Predictive Modeling: having a visualization of this process would be very helpful, even in the supp materials. This is otherwise very difficult to follow and to grasp what the authors did in the analysis.

We have now added a visualization that details the steps of the multivariate predictive modeling analyses in Supplemental Materials (Supplemental Figure 3).

10. Results: "We tested their generalizability in independent data using leave-one-site-out cross-validation in ABCD and 10-fold cross-validation in HCP" – why did the authors not use the same here? The rationale behind this is unclear.

As we now clarify in a parenthetical after this sentence (page 7), the HCP data were collected at a single study site, precluding the use of leave-one-site-out cross-validation.

11. Results: Figure 1 B: this figure is constraintuitive. Could the authors please add a dotted line as the average on the right plot, as is done on the left plot? Could these two plots be more similar so that the message behind the figure is clear?

We have now revised Figure 1B to make the plot of ABCD prediction accuracy across sites identical to the plot of HCP prediction accuracy across folds.

12. Results: 6: could this be driven by multiplying the weights by 1.5? What is the outcome of the analyses if you keep the weights as they are or change the multiplication factor?

As detailed in our response to comment #8, above, EEA estimates across different types of priors were nearly perfectly correlated with one another in both samples, indicating that prior choice was unlikely to affect the main findings of the study. Furthermore, the differences between samples that are reported in Section 6 all involve neural data rather than EEA estimates. As the priors were only used for modeling behavioral data and had no impact on analyses of the neural data, the results reported in Section 6 would be unaffected by prior choice.

13. Discussion: Perhaps the authors could discuss some limitations of their approach. Moreover, the discussion could be shedding light on more perspectives of the topic.

We have now added a section describing what we view as the study's main limitations and alternative perspectives to the Discussion on pages 20-21.

Reviewer #3

This is impactful work that sheds light over a general mechanism of cognitive performance and is thus likely to be of interest to a wide range of cognitive neuroscientists. I would recommend publication provided the authors can address a few concerns.

We thank the reviewer for their kind compliments about the manuscript and their constructive critiques that follow.

1. On paragraph #1 of the discussion: The authors state that "We demonstrate that neural responses to cognitive demand on the n-back can account for a large portion of the variance in EEA on the task and that this association is largely driven by neural responses across the FPN and DAN". The second part of this statement is difficult to sustain on the evidence presented. They did not compare, to the best of my understanding, the FPN and DAN with the five remaining Yeo networks. Thus, the association could be driven by one of the unexamined networks, but their statement seems to imply it is these two networks in particular that best explain the variance. If they wish to make this statement, I believe they should be comparing the variance explained by these two networks vs the rest of the parcellation.

We thank the reviewer for raising this excellent point and have now conducted more comprehensive analyses across the other Yeo networks to gauge the specificity of EEA's

associations with task-positive networks. As detailed in Section 3 of the Results, we now separately assess correlations between each Yeo network and EEA (Table 1, top panel). To our surprise, we found that the somatomotor network (SMN) had an association with EEA that was of similar size to those of the task-positive networks and that was consistent across both data sets. We also conducted additional analyses (page 9) demonstrating that the vast majority of the association between 2-0 contrast average brain network activations and EEA can be attributed to just these three networks (FPN, DAN, and SMN), and we therefore now include the SMN in all subsequent analyses. Curiously, these analyses suggest that the character of the SMN's association with EEA differs from that of the task-positive networks: the SMN is generally deactivated during task performance and individuals with lower EEA show similar levels of SMN deactivation across n-back conditions while those with higher EEA show greater deactivation in the 2-back, relative to 0-back, condition (Results Sections 4 and 5). We now discuss this new set of findings throughout the discussion, but especially on pages 18-19. We appreciate that the reviewer raised this point, as we believe these new analyses both reinforce the importance of task-positive networks' contributions to EEA and identify another interesting and divergent, if somewhat unexpected, pattern of findings in the SMN.

2. In the same paragraph, the authors state “Indeed, FPN and DAN activation measures drawn from only single levels of cognitive demand showed systematically weaker associations with EEA than measures of differences in FPN and DAN activation across high- and low-demand levels.” However, to the best of my understanding they did not perform a direct comparison of strength of association between one vs multiple conditions, nor of the amount of variance explained to sustain this statement. Near the middle of page 9 the authors mention that variance explained is higher in the between-conditions contrast relative to within-conditions, but the % numbers presented within brackets are quite close to each other and we do not know whether these were statistically significantly different. It is difficult to know if a difference of 2-5% points is relevant. Carrying out formal statistical testing would allow the reader to reach their own conclusion. As a formatting issue, it would be better to present the percentages of variance explained in Table 1.

We apologize that our previous reporting of comparisons between the strength of EEA associations with 2-0 activation differences versus single-condition activation estimates was difficult to interpret and did not highlight formal statistical evidence. We have now made two changes to address these issues. Firstly, rather than convert the values in Table 1 to percentages of variance explained, we now discuss the relevant comparisons using r -values and their 95% confidence intervals (CIs) because these are the effect size estimates that we use throughout the rest of the analyses. We realized that transitioning to variance explained for this specific analysis creates a disconnect with the rest of the Results section that is likely to be needlessly confusing for readers. Secondly, we now provide formal statistical evidence by comparing the 95% CIs for the r of 2-0 activation difference associations with EEA and the corresponding r s of single-condition associations. Non-overlapping 95% CIs indicate a clear statistically significant difference at the $p < 0.05$ significance threshold while overlapping 95% CIs do not necessarily indicate that a statistically significant difference does not exist between the values (Cumming, 2009, *Statistics in Medicine*). Therefore, we interpret all non-overlapping CIs as statistically significant differences and, for comparisons of values with overlapping CIs, we use clustered bootstrapping to estimate the 95% CI for the difference in r -values and assessed whether this difference CI overlaps with 0.

As is now detailed on pages 11-12 in Results Section 4, most comparisons between the EEA associations of 2-0 activation differences and single-condition estimates suggested that the 2-0 difference associations were clearly larger, as indicated by non-overlapping CIs. In one exception for the DAN in ABCD, the DAN 2-0 difference association ($r = 0.22$, $CI = 0.19 - 0.24$) was numerically stronger than the association involving 2-back activation alone ($r = 0.17$, $CI = 0.14 - 0.21$), but there was substantial overlap between the 95% CIs. When we estimated the 95% CI for the absolute difference between the DAN 2-0 and DAN 2-back alone r -values in ABCD, we found that it overlapped slightly with 0 ($CI = 0.00 - 0.08$), suggesting weaker evidence for a difference in this case. In a second exception, again in ABCD, the CI for the SMN 2-0 difference association ($r = -0.30$, $CI = -0.27 - -0.32$) was almost entirely overlapping with that of the association involving 2-back activation alone ($r = -0.29$, $CI = -0.26 - -0.32$) and the CI for the difference between these r -values clearly overlapped with 0 ($CI = -0.5 - 0.02$). As we now discuss in more detail on page 18, these comparisons suggest clear evidence that FPN 2-0 differences are consistently more closely related to EEA than single-condition estimates, somewhat weaker evidence for the same pattern in the DAN, and evidence for this pattern in the SMN in the HCP data set, but not ABCD. We have also removed the “Indeed...” sentence the reviewer references in the first paragraph in order to allow for more nuanced discussion of this pattern of findings in the section on pages 17-18.

3. Point #4 on pp 7-9 seems mostly descriptive. The authors show the striking pattern of people with low activation on the 0-back and high activation on the 2-back condition having better performance on the task (or EEA), but they do not indicate how they assessed statistically whether this relationship is significant at the individual level. In point #5 they perform an analysis binning the participants in 10 bins sorted by EEA and demonstrate that, at this 'group' level the relationship seems to hold (as an aside, the authors should clarify what is the 'brain activation index'?). Their case would be strengthened if they could show this relationship is significant at the individual level, though. Are they able to identify these relationships in individual contrasts? Is a subtraction of activation in 2-0 positively (and significantly) associated with EEA for the top performers, but negatively with the bottom performers? Can you test difference scores, for example? Or incorporate EEA as a covariate? I believe carrying out analyses that demonstrate the existence of this relationship at the individual level would constitute irrefutable evidence for the conclusions this paper draws.

We appreciate the reviewer's excellent point about the importance of further characterizing the relation between brain network responses to cognitive demand and EEA at the within-person level and we agree that this would greatly strengthen support for the idea that flexible engagement of brain networks underlies EEA. However, we do not think that within-person analyses can be easily included in current manuscript because they would require addressing methodological barriers that would not be straightforward to overcome and that would substantially increase the scope of the study beyond its focus on characterizing brain network dynamics that relate to between-person differences in EEA. Specifically, given that within-person analyses must be conducted at the level of individual trials or blocks, we would expect estimates of within-person changes in both EEA and neural responses to be very noisy at this level given the reduced amount of data available. Overcoming these measurement barriers is, in fact, a key component of our ongoing research program. We believe that advances in hierarchical Bayesian joint modeling methods that estimate models of behavioral and neural data as being described by a single multivariate normal distribution (Turner et al., 2015, *Psychological Review*; Turner et al., 2017, *J. of Mathematical Psychology*; Innes et al., in press, *PsyArXiv*) are particularly promising, and we are actively working on

applying these models to large-scale brain network data during the n-back task. Therefore, we now discuss the lack of information about within-person associations as a limitation of the current manuscript and as a key focus of future research that uses joint modeling methodologies to better characterize within-subjects links between neural and behavioral data (page 20).

Regarding the reviewer's concern about the "activation index", we now clarify (Figure 4 caption, page 28 in the Methods) that this index is simply the mean of activation parameter estimates across all vertices in the network.

4. On page 7, point #3, the authors mention a 'Gordon parcellation'. It hasn't been mentioned before, and the rest of the text implies the authors used Yeo's parcellation. Is this a typo? Which parcellation was used? Can they elaborate on and add a reference for the Gordon parcellation if it's not?

The reviewer is correct that this was a typo and that the Yeo parcellation was indeed the only parcellation used for this study. We apologize for this error, which has now been corrected, and we thank the reviewer for catching it.

MINOR

• P14, The authors should be more explicit regarding how they modelled the N-back tasks. Did they have an implicit baseline? If so, what was it? Were the linear contrasts (e.g., 2-back vs 0-back) collapsing across all task conditions, or was it one for each condition which were later averaged?

As we now clarify on pages 24-25, the individual task conditions, which included separate conditions for 0-back and 2-back blocks and for each category of stimuli (HCP: faces, places, body parts, tools; ABCD: happy faces, neutral faces, fearful faces, places), were all modeled with separate regressors (e.g., 0-back places, 2-back places, 0-back tools, 2-back tools, etc.). The baseline was left unmodeled, resulting in an implicit baseline. Linear contrasts were constructed for the 0-back, 2-back, 2-back minus 0-back conditions of interest. Each contrast was made to average over all of the relevant 0-back and/or 2-back conditions.

• P15, when the authors state "Individual runs were considered good if they passed visual inspection", can they elaborate on what exactly they were checking for during visual inspection? I find it impressive that they visually inspected 5k+ subjects and would like to hear more about how thoroughly and under which criteria these data were inspected.

We now report our standard visual inspection procedures in greater detail on page 24 and clarify that visual inspection procedures were carried out by one of the authors as well as by a team trained undergraduate research assistants under their supervision.

• P15, EEA Estimation: Even though the tasks have been described in detail elsewhere, it would be good if the authors included a brief description of the tasks/conditions used for ease of reading.

We agree with the reviewer that describing the tasks in greater detail would be beneficial. Therefore, we now describe the design of both the HCP and ABCD tasks within the respective Methods sections on each sample on pages 22-23.

- *P16, Multivariate Predictive Modelling: What variables were submitted to dimensionality reduction in the PCR to obtain the components? Was it 3-dimensional contrast images (i.e., was the actual data the voxels of the contrast results per subject)? A more elaborate, detailed description would be good for subjects not acquainted with the method.*

We now indicate (page 27) that the features submitted to dimensionality reduction were indeed the vertex-level activation data from across the entire cortical surface for the 2-back minus 0-back contrast. We also now include a visualization outlining the steps of the multivariate predictive modeling process in Supplemental Figure 3.

- *P16, Network Activation Extractions: What do the authors mean with “the contrasts averaged within each network”? Did they average the activation of the 0-back and 2-back contrasts? What is the logic of this? Or do they mean they averaged the activation within each network of the Yeo7 parcellation?*
- *P16, Relations Among Network Metrics and EEA: What variables were actually correlated? Was it the v parameters of the DDM with mean activation value per network (for each condition/contrast)? How did the authors deal with the different task categories when producing these contrasts? This seems important to outline explicitly.*

We now clarify (page 28) that we averaged the activation of each contrast within each network of the Yeo 7-network parcellation by taking the mean of activation estimates across all vertices within each specific network. These average activation estimates were computed separately for the 0-back and 2-back contrasts (relative to implicit baseline) as well as for the 2-back minus 0-back contrast. Relations among these average activation values for each network and each contrast and relations of these activation values with EEA were then characterized in the subsequent analyses.

- *The authors could elaborate on how consensus maps work for readers that might be unfamiliar. What are they? How were they constructed? This information could be added to the methods.*

Thank you for this helpful suggestion. We now clarify in the Results on page 7 that these maps “indicate the relative importance of activation from each cortical surface area for predicting EEA”. We also describe how these maps are generated on page 28.

REVIEWERS' COMMENTS:

Reviewer #2 (Remarks to the Author):

Thanks to the authors for addressing my questions. I am satisfied with the author's responses to the issues raised in my initial review. The revised manuscript is now easier to follow after the incorporation of the feedback from the reviewers. I would like to recommend the authors for their following replies to reviewers comments to either also indicate the line numbers where the changes have been performed (in addition to the page numbers), or even highlight the changes in the manuscript - this would make it much easier to re-revise and evaluate the changes undertaken by the authors. This way, it is a time-consuming process to search through the manuscript to find the passages that were modified. Many thanks in advance.

Reviewer #3 (Remarks to the Author):

The authors have satisfactorily addressed all concerns I raised and I believe this has improved the quality of the work presented. I recommend publication.